# Current Characteristics Estimation of Si PV Modules Based on Artificial Neural Network Modeling

**DOI:** 10.3390/ma12183037

**Published:** 2019-09-19

**Authors:** Xiaobo Xu, Xiaocheng Zhang, Zhaowu Huang, Shaoyou Xie, Wenping Gu, Xiaoyan Wang, Lin Zhang, Zan Zhang

**Affiliations:** School of Electronic and Control Engineering, Chang’an University, Xi’an 710064, Chinawpgu@chd.edu.cn (W.G.); wxyswallow7907@163.com (X.W.); zhanglin_dk@chd.edu.cn (L.Z.); z.zhang@chd.edu.cn (Z.Z.)

**Keywords:** artificial neural network, PV module, current characteristics prediction

## Abstract

In the photovoltaic (PV) field, the outdoor evaluation of a PV system is quite complex, due to the variations of temperature and irradiance. In fact, the diagnosis of the PV modules is extremely required in order to maintain the optimum performance. In this paper, an artificial neural network (ANN) is proposed to build and train the model, and evaluate the PV module performance by mean bias error, mean square error and the regression analysis. We take temperature, irradiance and a specific voltage for input, and a specific current value for output, repeat several times in order to obtain an I-V curve. The main feature lies to the data-driven black-box method, with the ignorance of any analytical equations and hence the conventional five parameters (serial resistance, shunt resistance, non-ideal factor, reverse saturation current, and photon current). The ANN is able to predict the I-V curves of the Si PV module at arbitrary irradiance and temperature. Finally, the proposed algorithm has proved to be valid in terms of comparison with the testing dataset.

## 1. Introduction

Manufacturers provide standard reporting conditions (SRC) or standard test conditions (STC) ratings for photovoltaic (PV) components. The solar simulator is mainly used under laboratory conditions. These conditions include the intensity of 1000 W/m^2^, the spectrum distribution conforming to AM1.5 spectrum, and the temperature of PV modules 25 ±1 °C. However, in practical cases, these conditions rarely appear, and the adequacy and applicability of PV modules under STC are a controversial issue. In fact, energy collection optimization of PV modules based on SRC efficiency is tough and misleading for actual weather conditions, therefore, proper characterization of the electrical performance (I-V curve) of PV modules is a basic requirement of PV engineering [1].

Although the outdoor behavior of PV modules can be predicted by algebraic or numerical methods, PV system engineering tends to adopt the former algorithms by ignoring some second-order effects: wind speed, shunt resistance of cells, parasitic capacitance, spectral effects, nonlinearity under low illumination, etc. [2,3,4,5]. The most widely used single diode analytical model is based on the equivalent circuit which is composed of a current source, a diode, a series resistance, and a shunt resistance. A large number of models have been reported before, with 3 [6], 4 [7,8], or 5 [9,10,11] parameters. Most methods use the I-V characteristics under illuminated conditions, and better performance lies in fewer approximations and more data. Therefore, the methods derived from two or more I-V curves under different levels of illumination, without assuming the ideality factor to be 1, or the shunt resistance to be infinite, show the best accuracy [12,13]. In this paper, Khan’s algorithm [4] which extracts the parameters under arbitrary irradiance, and assumes them to be variable, is chosen to be compared with our proposed model. 

Apparently, the performance of analytical methods depends on the parameter accuracy, while the data-driven artificial neural network (ANN) algorithm abandons the predominated parameters and equivalent circuit, and is able to build the PV model from historical data with no assumption. ANN has been widely applied in the PV field [14], such as the estimation and prediction of global solar irradiance data [15,16], maximum power point tracking of PV modules [17,18,19], and performance prediction of the PV module using electrical equivalent model [20]. In this paper, ANN is used to predict the I-V curve of single crystal silicon modules under different irradiances and temperatures without any parameters, and the prediction accuracy is proved to be better than the parameter based method. The aim is to predict the electrical performance of PV modules under desired conditions without spending many resources to do a lot of monitoring, so as to reduce the energy waste of PV engineering in practical applications. The model can generate I-V curves of silicon crystal PV modules under any irradiance and ambient temperature, ranging from 100 to 1300 W/m^2^ and 10 to 36 °C, which is validated by the performance data of the PV module installed in the city of Cocoa, Florida, afforded by the National Renewable Energy Laboratory (NREL).

The paper is structured as follows. Firstly, the technical characteristics of the experimental device used in this paper and the related data sets are introduced. The third part describes the structure and prediction results of the ANN model. The fourth part demonstrates the estimation accuracy and discusses the prediction results. In the last part, the conclusion of this paper is summarized.

## 2. Experimental Device

The performance and technical characteristics of PV modules depend on the climatic conditions of the installation location, probably leading to overestimation or underestimation of energy production under actual working conditions. In fact, compared with the output under standard test conditions, the energy output proposed in the literature may be 40% higher than the actual production [21]. It should be recognized that the correctness and accuracy of the characteristic parameters of PV modules under actual operating conditions are of great significance to the determination of their performance. Therefore, the NREL has set up an outdoor test-bed in the city of Cocoa, Florida, USA to provide realistic data for evaluation, thus providing a realistic basis for validating the neural network model methods that are currently used.

The experimental system is located in Cocoa, Florida, on the prominent peninsula off the southeast coast. The installation inclination angle of the PV module is 28.5 degrees, facing south. Figure 1 shows the deployment of the PV modules and equipment.

The NREL provides users manual describing performance data for flat-panel PV modules installed in Cocoa, Florida. The data includes current-voltage (I-V) curves of the PV modules of all flat panel PV technologies and integrated data sets of related meteorological data for about a year. The data includes various irradiance and temperature conditions representing each season at each location. The public data is intended to facilitate validation of existing models to predict the performance of PV modules and to develop new and improved models [22].

## 3. Construction of the Model

Among many neural network models, the supervisory model is the most widely used model in machine learning and it is also the most effective and easy-to-use neural network [23]. Multi-layer perceptron (MLP) is the most common way to implement the supervisory model [24].

A combination of MLP and the gradient descent method results in a very effective algorithm, known as the back propagation (BP) algorithm [25,26,27,28,29]. The main idea of the gradient descent method is to make the weight of each node move into the negative direction of the loss function gradient and make the network adjust the weight value of each node by itself. Input vectors and the corresponding output vectors are used in the training of neural networks. Finally, the neural networks can be approximated as a non-linear function that can associate input vectors with specific output vectors. The output vector achieves the function of “prediction”, enabling us to obtain more accurate input/output results in a wide range only by training some ranges of input/output networks.

In this study, the advantages of MLP are utilized without understanding the internal structure of the system. If an I-V curve is observed, we have “*Curve* (*V*-*I*) = *f*(*T_C_*, *G*)”, with *T_C_* the temperature (°C) and *G* the irradiance (W/m^2^) of the PV module. From the measured data in Figure 2, different temperatures and irradiances correspond to different I-V curves. Therefore, we can assume that there is a functional relationship between *I*, *V*, *T* and *G*, the problem is simplified into a non-linear functional calculation. The BP network does well in fitting non-linear functions, and it is able to find the law between the output I-V curve and temperature and irradiance in a pile of training data points.

As it is difficult to make a set of temperatures and irradiances corresponding to (I, V) data pairs, the function *Curve*(*V*-*I*) = *f*(*T_C_*,*G*) is adjusted to *I* = *f*(*T_C_*,*G*,*V*), therefore, the BP network only needs to find a set of rules between temperature, irradiance, voltage, and current. At the same temperature and irradiance, multiple sets of voltage and current values can be obtained and the I-V curve can be drawn.

The structure of the neural network consists of three layers, as shown in Figure 3. The first layer (the input layer) has three neuron nodes (*T_C_*, *G*, *V*), the second layer (the hidden layer) has three nodes, the last layer (the output layer) has only one node, denoted as the output current value. The difficulty is that the structure of the MLP network is mainly determined by experience. There is no effective formula to calculate the structure parameters of the MLP network for different situations [30]. 

The concrete steps of model construction are as follows:

### 3.1. Step 1, Obtain Actual I-V Curves

In order to ensure the practicability of the prediction model, the I-V curve must be measured under real solar irradiance conditions. In order to obtain the true I-V curve, we used the data of xSi12922 from the NREL.

This paper uses a series of data of PV components at the Cocoa site from 21 January 2011 to 4 March 2012. The experimental PV components are purchased and installed by the NREL. The relevant tests have been carried out, and all the data has been filtered. Some suspicious data has been discarded, which ensures the authenticity of the experimental data to the maximum extent. 

### 3.2. Step 2, Select the Appropriate Training Set

In the training process of MLP, the selection of the training set is vital. Only when it fully represents the module behavior, the network can train reasonable MLP.

In the initial stage, less data pairs are selected for training, and the gradient and specificity of the data pairs are not paid attention to, resulting in unsatisfactory results. The data points used at the beginning are with (25 °C, 917.68 W/m^2^), (25 °C, 958.39 W/m^2^), (25 °C, 957.11 W/m^2^), (25 °C, 951.12 W/m^2^), (25 °C, 1000 W/m^2^), (25 °C, 927.86 W/m^2^) and (25 °C, 930.58 W/m^2^). The last group is the testing data set.

Afterwards, the reasons for large deviations are analyzed, the relevant information is consulted, and the training set is re-selected according to the database. Because the data set with a certain gradient can not be found at the edge temperature (such as <10 °C) or the edge illumination intensity (such as <100 W/m^2^), the training set is adjusted within the specified range. Table 1 is the temperatures and illumination intensities corresponding to the final selected training set, and the I-V curve corresponding to each temperature and intensity. In this case, the BP neural network could generate the I-V curve under any operating condition.

### 3.3. Step 3, Generation of the I-V Curve

With a well trained MLP, the predictive current characteristics of arbitrary outdoor condition from the testing data set is selected and verified by the actual one. The model constructed in this experiment can be predicted in the range of 10 °C~35 °C and illumination intensity in the range of 100 W/m^2^~1200 W/m^2^. It should be noted that when predicting the edge temperature or edge illumination intensity, the limitation of the training set and the error of the actual measurement environment are large, resulting in unreliable estimations.

The final model is tested comprehensively, with the temperatures and illumination intensities selected randomly, leading to prediction results (I-V curves) obtained in Figure 4. It can be seen that the mean square error (MSE) of three cases (10.1 °C and 595.9 W/m^2^, 16 °C and 483.9 W/m^2^, 28.3 °C 895.6 W/m^2^) are with the order of 10^−3^, and MSE of 35 °C and 950.5 W/m^2^ is 10^−2^, for both predictions. The predictions coincide well with the measured data, showing the validity of our algorithm. 

## 4. Analysis of the MLP Prediction Results

For the I-V curve generated by MLP, the fitting degree of the curve is high from Figure 4, but in order to describe the accuracy of the predictive I-V curve more carefully, it is essential to introduce more parameters to estimate the results.

The mean bias error (MBE), MSE, and the coefficient of determination (R^2^) between the actual curve and the curve obtained by MLP should be calculated [31,32]. The network response (A) and the corresponding target (T) of the regression analysis should be analyzed so as to study the network response in detail. MBE, MSE and R^2^ are defined as follows:(1)$$MBE=\frac{1}{N}\sum_{i=1}^{N}\left({\hat{y}}_{i}-y_{i}\right)$$
(2)$$MSE=\frac{1}{N}\sum_{i=1}^{N}{\left({\hat{y}}_{i}-y_{i}\right)}^{2}$$
(3)$$R^{2}=\frac{{\left(N\sum_{i=1}^{N}{\hat{y}}_{i}y_{i}-\sum_{i=1}^{N}{\hat{y}}_{i}\sum_{i=1}^{N}y_{i}\right)}^{2}}{\left(N\sum_{i=1}^{N}{\hat{y}}_{i}{}^{2}-{\left(\sum_{i=1}^{N}{\hat{y}}_{i}\right)}^{2}\right)\left(N\sum_{i=1}^{N}y_{i}^{2}-{\left(\sum_{i=1}^{N}y_{i}\right)}^{2}\right)}$$
where ${\hat{y}}_{i}$ (*i* = 1,2,…,*n*) is the predicted value of the sample; $y_{i}$(*i* = 1,2,…,*n*) is the actual value of the sample; *N* is the number of samples.

Explanation: 1. MSE (MBE) can evaluate the change degree of the data. The smaller the value of MSE (absolute value of MBE), the better the accuracy of the predictive model to describe the experimental data. 2. The coefficient of determination is within the range of [0, 1], the closer to 1, the better the performance of the model. On the contrary, a value closer to 0 causes a worse result.

Secondly, in order to further test the accuracy of the prediction results, the linear regression analysis of the predicted current value and the actual one is carried out. The regression function is: ${\hat{y}}_{i}=\alpha +\beta x_{i}$, where ${\hat{y}}_{i}$ is the actual value, $x_{i}$ is the predicted value, α and β correspond to the *y* intercept and the slope of the best linear regression related to the network output, respectively. The ideal regression function is: ${\hat{y}}_{i}=x_{i}$. That means if we have a perfect fit (the output is exactly equal to the target), the slope is 1 and the *y* intercept is 0.

Figure 5 illustrates the graphical outputs provided by the regression analysis. The linear fitting of the MLP prediction results is represented by red dots, that of the actual values is denoted by blue asterisks, and the perfect fitting (output equals target) is demonstrated by solid lines. In this case, it is difficult to distinguish the linear fitting line of the MLP prediction results from the actual fitting line, as both lines nearly coincide with each other. Table 2 illustrates the fitting accuracy of I-V curves between measured data of outdoor conditions and those generated by our model for the testing data set. The intercept (α) of the regression analysis curve approaches 0, the slope (β) and 1 are covering, and R^2^ is expected to get close to 1. These parameters reveal that the I-V curves generated by MLP and the actual ones are nearly overlapping. It is important to point out the ability of MLP to generate the I-V curve. Any I-V curve can be generated to ensure that the result approaches the actual one. It is noteworthy that this method can generate I-V curves of PV modules under the conditions of ambient temperature ranging from 10 °C~35 °C and illumination intensity ranging from 100 W/m^2^~1200 W/m^2^.

In order to further estimate the predictive performance of the model, we choose three days (June 8, 18, and 29, 2011 ) from the original I-V curve test samples provided by the NREL, keep the voltage fixed to be 1 V, extract the corresponding current value, with the temperatures and illumination intensities corresponding to different time of the days as the input. The predicted current is compared with the actual one in a graph. The performance of the constructed neural network model is evaluated by the fitness of the two curves. Figure 6 shows that the predicted current value is very close to the actual one at various time of three days, which proves the accuracy of MLP model prediction.

Finally, the ANN method is compared to the parameter based analytical method [4], shown in Table 3. The I-V data is selected randomly from the testing data set. Apparently, the ANN method is highly in accordance with the measured ones at different operation conditions, whereas the parameter based model obviously deviates with the measured data. 

## 5. Conclusions

In this paper, the ANN algorithm combining MLP with the gradient descent method is applied to the prediction of the current characteristics of a Si PV module with arbitrary outdoor condition. The main feature of our algorithm is the ignorance of any predominated parameter, resulting in higher coincidence between the predicted data and the measured one. Furthermore, the model is even valid for extreme weather conditions with a high fitting degree. The estimation accuracy has been widely investigated by MSE, MBE and the regression analysis, with numerous testing data set originated from the NREL repository. 

The core superiority of the proposed method is because of the black-box data-driven property, and the specific explanation is that with the ANN trained and build from abundant data, where the weight factors and biases are calculated automatically, we are able to estimate the output current directly from a new temperature and a new irradiance, without any assumption introduced. It doesn’t need any initial guesses and extraction of any critical electrical parameter (like the series resistance or the shunt resistance). The comparison of our method with the parameter-based one reveals that our algorithm is better with a far smaller MSE. In the future, we will consider applying deep-learning methods based on convolutional Neural Networks to PV modules manufactured with all technologies and materials.

## Figures and Tables

**Figure 1 materials-12-03037-f001:**
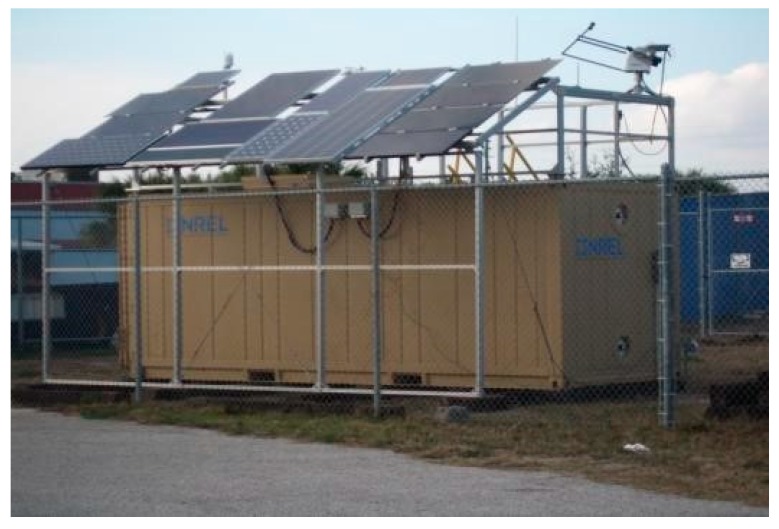
Deployment of PV modules and equipment in Cocoa, Florida.

**Figure 2 materials-12-03037-f002:**
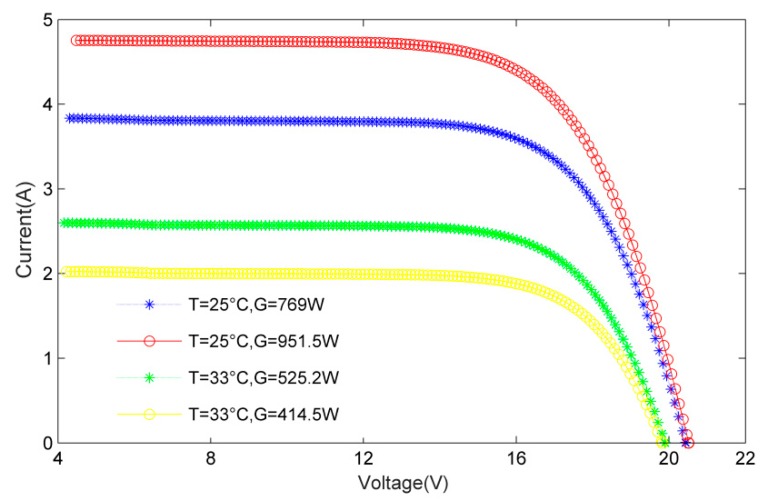
Current characteristics for different T and G.

**Figure 3 materials-12-03037-f003:**
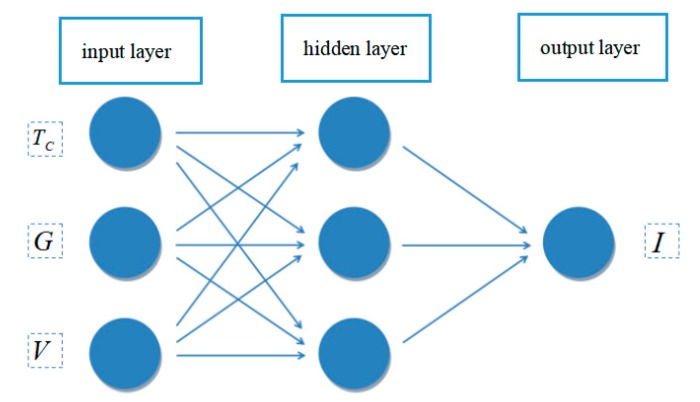
MLP structure of the neural network.

**Figure 4 materials-12-03037-f004:**
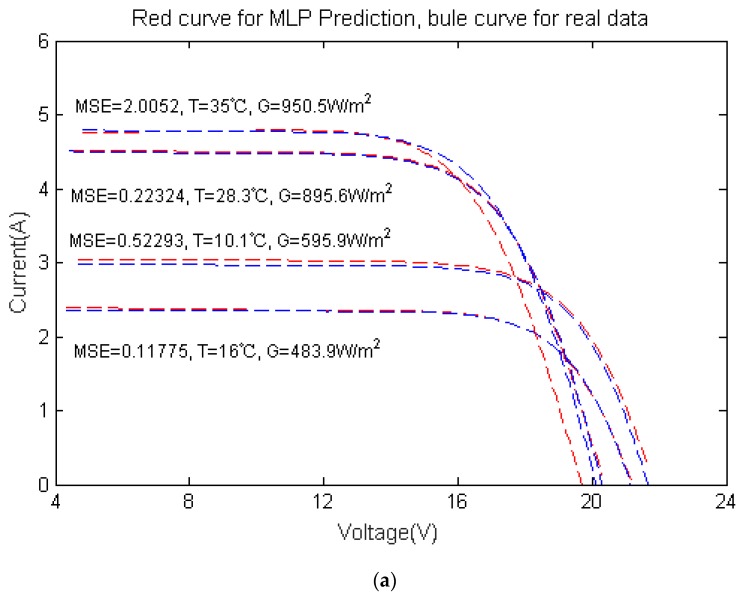
Prediction results when the temperature is 10.1 °C, 16 °C, 28.3 °C, and 35 °C, with the corresponding illumination intensity 595.9 W/m^2^, 483.9 W/m^2^, 895.6 W/m^2^, and 950.5 W/m^2^, respectively; (**a**) first prediction results; (**b**) second prediction results.

**Figure 5 materials-12-03037-f005:**
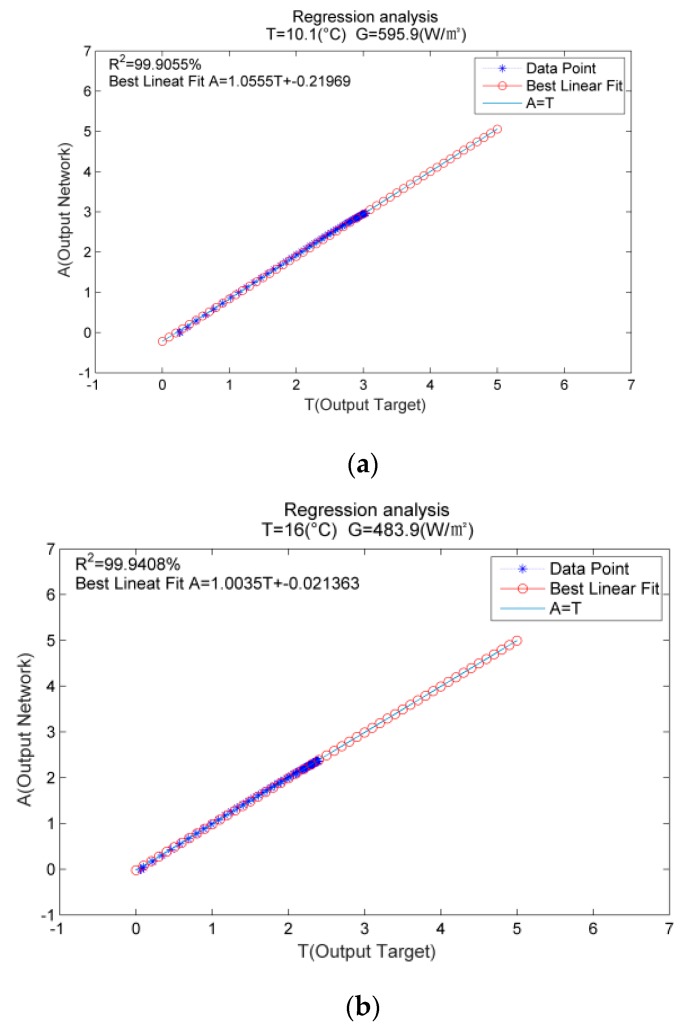
Regression analysis results corresponding to different temperatures and irradiances (**a**) T = 10.1 °C, G = 595.9 W/m^2^; (**b**) T = 16 °C, G = 483.9 W/m^2^; (**c**) T = 28.3 °C, G = 895.6 W/m^2^; (**d**) T = 35 °C, G = 950.5 W/m^2^.

**Figure 6 materials-12-03037-f006:**
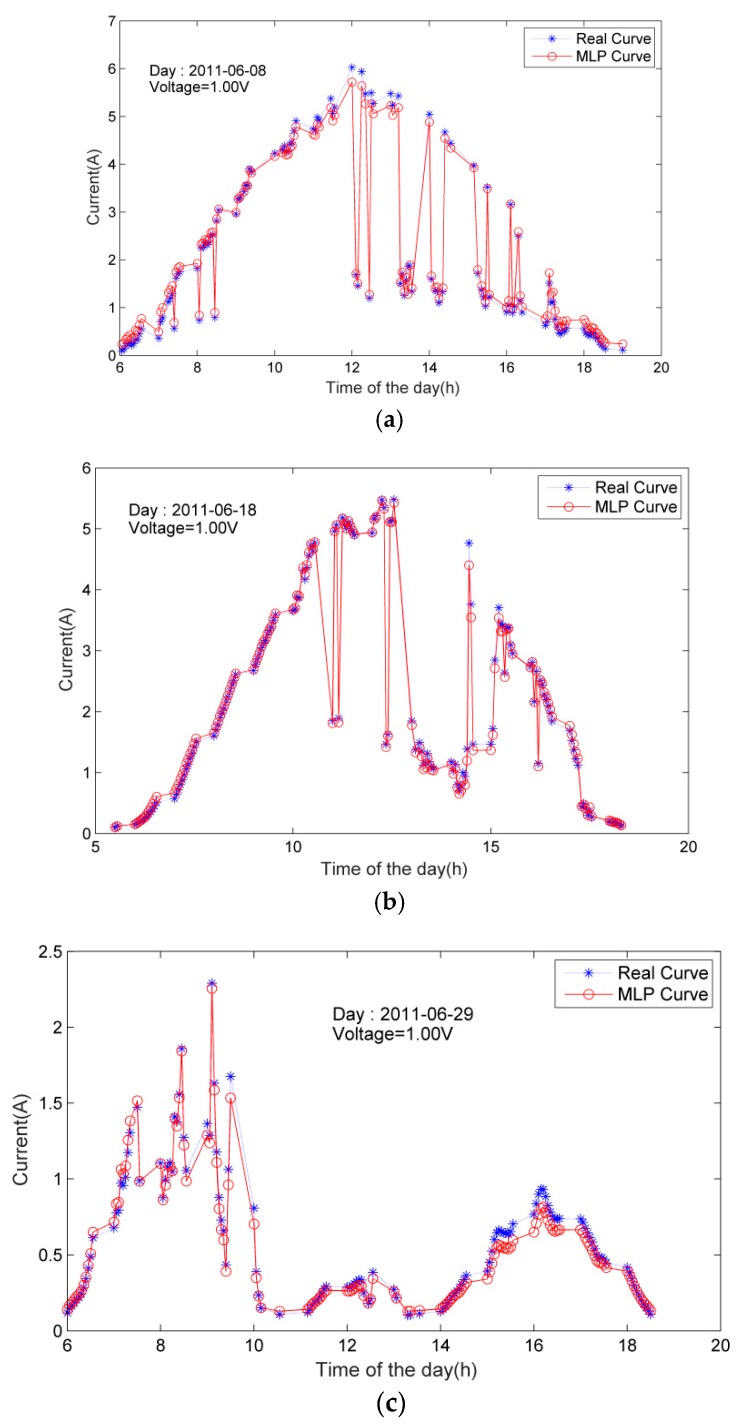
Comparison of the performance prediction results of the PV module on (**a**) 8 June 2011; (**b**) 18 June 2011; (**c**) 29 June 2011.

**Table 1 materials-12-03037-t001:** Irradiance (*G*) and ambient temperature (*T_C_*) of the PV module for training MLP to obtain I-V curves.

G (W/m^2^)	T_c_ (°C)	G (W/m^2^)	T_c_ (°C)	G (W/m^2^)	T_c_ (°C)	G (W/m^2^)	T_c_ (°C)
100.3	12.9	102.1	20.0	109.8	25.0	104.0	33.0
346.7	12.9	352.0	20.0	349.9	25.0	324.7	33.0
608.5	12.7	604.0	20.0	601.9	25.0	601.1	32.3
873.3	12.9	847.9	20.1	847.4	25.0	845.7	34.4
1105.0	12.5	1052.3	20.0	1088.2	25.0	1097.2	34.6

**Table 2 materials-12-03037-t002:** MSE, MBE, and parameters of regression analysis for measured and predicted curves.

xSi12922					
G (W/m^2^)	T_c_ (°C)	MSE (%)	MBE (%)	Best linear fit: A = βT + α	R^2^(%)
207.9	21.6	0.14357	−0.0081567	A = (0.95983)T + (0.044092)	99.3498
491.7	15.5	0.33843	0.036869	A = (1.06340)T + (0.168920)	99.8593
922.9	23.8	0.36043	0.0094	A = (0.97613)T + (0.120550)	99.8593
639.1	25.0	0.33031	0.035184	A = (1.01750)T + (−0.09513)	99.9086
384.7	18.3	1.59730	−0.29256	A = (1.00290)T + (0.253630)	99.5170
1059.6	30.3	0.48341	−0.025697	A = (0.97268)T + (0.087804)	99.8283
443.7	24.3	0.34652	−2.4395	A = (1.05980)T + (−0.15873)	99.9102
895.6	28.3	0.24494	0.17058	A = (1.01460)T + (0.033725)	99.9439
380.3	25.7	0.54364	0.0438	A = (0.86450)T + (0.164720)	99.4472
565.2	22.3	0.80404	0.03101	A = (0.94354)T + (0.022098)	99.8563
768.6	31.6	0.83609	0.0305	A = (1.07590)T + (0.123480)	99.9921
922.9	31.8	0.88071	0.0541	A = (0.92050)T + (0.386470)	99.9315
930.2	33.2	0.75781	0.1589	A = (1.05340)T + (0.313870)	99.9730
483.9	16.0	0.15812	0.0076	A = (0.97426)T + (0.041104)	99.8693
1066.7	35.0	1.79780	0.1181	A = (0.86818)T + (0.743890)	99.9504
650.1	30.0	1.81190	0.03271	A = (1.50320)T + (1.44840)	99.6288
951.5	25.0	0.76103	0.03312	A = (1.07101)T + (0.37733)	99.9840
1000.8	32.5	0.59843	0.0929	A = (0.96008)T + (0.23463)	99.9368
525.2	33.0	2.72460	0.00837	A = (0.63929)T + (0.95310)	99.2597
950.5	35.0	1.46910	0.12831	A = (0.89510)T + (0.50351)	99.6683

**Table 3 materials-12-03037-t003:** The MSE between the measured and the predicted data for the ANN method and Khan’s analytical one.

G (W/m^2^)	T_c_ (°C)	ANN MSE (%)	Khan MSE (%)
425.5	38.2	0.43587	3.38521
1032.6	47.1	0.35621	1.82391
281.4	34.7	0.57120	5.93176
106.4	24.8	0.19348	9.12813

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
