# Peer review of "Current Characteristics Estimation of Si PV Modules Based on Artificial Neural Network Modeling"

_materials, 2019, doi:10.3390/ma12183037_

Round 1
Reviewer 1 Report
Manuscript presents a method based on ANN to predict the IV curve of Si PV module under different irradiance and temperature conditions.
The research topic is found interesting and relevant to the journal scope, however the paper is not suitable for the publication.
The manuscript is not well organized and presents a fragmented literature review. The background information as well as the introduction section are not adequate to understand the aims and objectives of the study. What is the novelty of the present study? The section 2 and 4 are not relevant in the present form. Other issues to address:
The reference [1] is missing; Too many references in bulk in the raw 33; Check the axes title in Figure 2; Check the formatting of the Table 2.The authors should perform a detailed analysis of the forecasting accuracy. Please check the paper “Forecasting of PV Power Generation using weather input data‐preprocessing techniques, Energy Procedia, Volume 126, 2017, Pages 651-658, https://doi.org/10.1016/j.egypro.2017.08.293”
The authors should quantify the potential improvement by using the method-based ANN in comparison with parameter methods. See the paper "Solar irradiance and temperature influence on the photovoltaic cell equivalent-circuit models, Solar Energy, Volume 188, 2019, Pages 1102-1110, https://doi.org/10.1016/j.solener.2019.07.005".
Author Response
1. The manuscript is not well organized and presents a fragmented literature review. The background information as well as the introduction section are not adequate to understand the aims and objectives of the study. What is the novelty of the present study? The section 2 and 4 are not relevant in the present form. Other issues to address:
The reference [1] is missing; Too many references in bulk in the raw 33; Check the axes title in Figure 2; Check the formatting of the Table 2.
Response: In Section 1, the background has been rewritten, and the novelty has been clarified; Section 2 has been deleted. All the references has been edited, Figure 2 has been revised. The format of Table 2 has been adjusted.
2. The authors should perform a detailed analysis of the forecasting accuracy. Please check the paper “Forecasting of PV Power Generation using weather input data‐preprocessing techniques, Energy Procedia, Volume 126, 2017, Pages 651-658, https://doi.org/10.1016/j.egypro.2017.08.293”
Response: According to the related paper analysis, mean bias error (MBE), mean square error (MSE), the linear fit and the coefficient of determination are normally investigated to show the accuracy. Such as “Characterisation of PV CIS module by artificial neural networks. A comparative study with other methods”, Renewable Energy, volume 35, 2010, Pages, 973-980.
Therefore the mean square error analysis has been added in Table 2.
3. The authors should quantify the potential improvement by using the method-based ANN in comparison with parameter methods. See the paper "Solar irradiance and temperature influence on the photovoltaic cell equivalent-circuit models, Solar Energy, Volume 188, 2019, Pages 1102-1110, https://doi.org/10.1016/j.solener.2019.07.005".
Response: A single-diode analytical method based on two or more I-V curves has been adopted, to compare with the ANN based model, listed in Table 3. The result shows that our model agrees better with the measured data.
Reviewer 2 Report
Comments:
This article deals with a model (using machine learning approach) to evaluate the PV module. It is claimed that the I-V characteristics of Si-PV module can be predicted. The authors have discussed the construction and validation of the model.
The comments on this article are listed below.
I am confused on some abbreviations; ANN; they have mentioned artificial neutral network (line 11), artificial neural network (line 36 and 50), etc. It has mentioned I-V curves (line 14, line 17), V-I curves also have mentioned many times. Is there any reason to alter the notation? Please check the abbreviations and other notations thoroughly. The axis of Fig. 2 looks error. To establish the model, how the authors did chose the input parameters? How do you consider the azimuthal angle to account solar radiation? Are the hidden layers also fixed? Please discuss the first prediction and second prediction. It would be better if Fig. 4 - Fig. 7 are merged.Author Response
I am confused on some abbreviations; ANN; they have mentioned artificial neutral network (line 11), artificial neural network (line 36 and 50), etc. It has mentioned I-V curves (line 14, line 17), V-I curves also have mentioned many times. Is there any reason to alter the notation? Please check the abbreviations and other notations thoroughly. The axis of Fig. 2 looks error. To establish the model, how the authors did chose the input parameters? How do you consider the azimuthal angle to account solar radiation? Are the hidden layers also fixed? Please discuss the first prediction and second prediction. It would be better if Fig. 4 - Fig. 7 are merged.
Response:
The “neutral” have been replaced with “neural”, the current characteristics has been denoted as “I-V” curve in the whole manuscript.
Fig 2 has been modified.
In principle, we can choose the input data randomly, but it is thought to be more convincing if the data covers more temperature and irradiance ranges.
The azimuthal angle and the hidden layer are calibrated by NREL, we can see the detail in “
User’s Manual for Data for Validating Models for PV Module Performance” by W. Marion, A. Anderberg, C. Deline, S. Glick, M. Muller, G. Perrin, J. Rodriguez, S. Rummel, K. Terwilliger, and T.J. Silverman, (https://www.nrel.gov/docs/fy14osti/61610.pdf)
The first and the second predictions has been discussed, Fig4-7 have been merged.
Round 2
Reviewer 1 Report
The authors addressed every comments. The paper can be accepted in the present form.
Reviewer 2 Report
The authors have addressed the comments on the manuscript.
I recommend it for publication.
Cheers